# Influence of Beetroot Juice Ingestion on Neuromuscular Performance on Semi-Professional Female Rugby Players: A Randomized, Double-Blind, Placebo-Controlled Study

**DOI:** 10.3390/foods11223614

**Published:** 2022-11-12

**Authors:** Álvaro López-Samanes, Juan José Ramos-Álvarez, Francisco Miguel-Tobal, Sofía Gaos, Pablo Jodra, Raquel Arranz-Muñoz, Raúl Domínguez, Juan José Montoya

**Affiliations:** 1Exercise Physiology Group, Faculty of Health Sciences, School of Physiotherapy, Universidad Francisco de Vitoria, 28223 Madrid, Spain; 2Faculty of Medicine, School of Sport Medicine, Madrid Complutense University, 28040 Madrid, Spain; 3Faculty of Education Sciences, University of Alcalá, 19001 Alcala de Henares, Spain; 4Departamento de Motricidad Humana y Rendimiento Deportivo, University of Seville, 41013 Seville, Spain; 5Studies Research Group in Neuromuscular Responses (GEPREN), University of Lavras, Lavras 37200-000, Brazil

**Keywords:** female athletes, endurance, ergogenic aids, nitrates, sport nutrition

## Abstract

Purpose: Beetroot juice (BRJ) is considered an ergogenic aid with good to strong evidence for improving human performance in sport modalities with similar demands to rugby. However, most of the studies were realized in male athletes with limited evidence in female athletes. Thus, the aim of this study was to explore the acute ingestion of BRJ in female rugby players. Methods: Fourteen semi-professional female rugby players (25.0 ± 3.7 years) belonging to a team from the First Spanish Female Rugby Division participated in this study. Participants were randomly divided into two groups that realized a neuromuscular battery after BRJ (140mL, 12.8 mmol NO_3_^−^) or placebo (PLAC, 140 mL, 0.08 mmol NO_3_^−^) ingestion on two different days separated by one week between protocols. The neuromuscular test battery consisted of a countermovement jump (CMJ), isometric handgrip strength (i.e., dominant), 10-m and 30-m sprint, agility *t*-test and Bronco test. Afterwards, participants reported a rate of perception scale (6–20 points) and side effects questionnaire associated with BRJ or PLAC ingestion. Results: Statistically significant improvements were observed in CMJ (7.7%; *p* = 0.029; ES = 0.62), while no differences were reported in dominant isometric handgrip strength (−1.7%; *p* = 0.274; ES = −0.20); 10-m and 30-m sprint (0.5–0.8%; *p* = 0.441–0.588; ES = 0.03–0.18); modified agility *t*-test (−0.6%; *p* = 0.503; ES = −0.12) and Bronco test (1.94%; *p* = 0.459; ES = 0.16). Conclusions: BRJ ingestion could improve neuromuscular performance in the CMJ test, while no differences in sprint (10-m and 30-m sprint test), agility, isometric handgrip strength and endurance performance (i.e., Bronco test) were reported

## 1. Introduction

The rugby union is one of the most popular collision sports, with over 8.5 million registered players over the world [1]. Currently, the number of female rugby players is increasing significantly. However, the number of studies in the rugby union are limited concerning female compared to male rugby players [2]. Rugby is a team sport that combines high intensity actions (i.e., sprinting, running, tackling, rucking, scrummaging and mauling) with actions at a lower intensity (i.e., jogging and walking or standing) [3]. An official match using GPS technology has reported that female rugby players cover ~5.8 km with the predominance of this distance in low intensity actions, such as standing or walking (42.7%) and jogging (35%), followed by running at low (9.7%) and moderate (9.5%) intensity and actions of high intensity, such as running at high intensity (1.8%) or sprinting (1.2%). Nevertheless, collisions and specific actions in rugby provoked the rugby players to present heart rates higher >90% of the maximum during the match [3]. In addition, an increment of the number of accelerations and decelerations in the latest year were shown to be related to an increase in physical demand for these athletes [2]. Female rugby players should develop their anthropometric and physical qualities (i.e., muscle and power strength, endurance capacity, linear speed and change of direction) for high performance [4].

With the aim of enhancing performance, athletes ingest sport supplements (SS) [5]. In fact, the International Olympic Committee (IOC) has classified SS, based on the level of scientific evidence, and identified a limited number of SS that could exert ergogenic effects [6]. The limited list of SS considered ergogenic aids with a high level of scientific evidence includes nitrate (NO_3_^−^) supplementation. However, NO_3_^−^ does not have any biological effect. After ingestion, NO_3_^−^ is reduced partially to nitrite (NO_2_^−^) in the mouth by NO_3_^−^ reductase produced by microorganisms [7]. Later, in the stomach and systemic circulation, NO_2_^−^ is reduced to nitric oxide (NO) [8]. Via NO, NO_3_^−^ supplementation induces several ergogenic effects, including an improvement in vasodilation [9], increasing muscular blood flow [10], metabolic response [11] and muscle force contraction. Beetroot juice (BRJ) is an enriched nutritional source of NO_3_^−^ and was reported to induce higher ergogenic effects than NO_3_^−^ salts supplementation [12]. This advantage of BRJ compared to NO_3_^−^ salt supplements could be mediated by other compounds that favor the conversion of NO_2_^−^ to bioactive NO [12].

The ergogenic effect of BRJ/NO_3_^−^ supplementation was associated with endurance performance [13]) with an enhanced economy (reduction in VO_2_ at the same intensity [14]) [15] and time-to-exhaustion [16,17]; but these ergogenic effects are not only for endurance performance because BRJ is an effective SS for enhancing performance in intermittent high-intensity exercise efforts [18] and explosive efforts [19]. Nevertheless, the assessment of the ergogenic effect of BRJ in the female athlete population is scarce. In this way, a recently published systematic review on this topic showed that female participants represented only ~7% of the sample size [19]. In addition, only seven studies have analyzed the effect of BRJ on a sample composed exclusively of female athletes [20,21,22,23,24,25]. Among them, two studies recruited female water polo players [22] and swimmers [24], while the other three studies assessed the combined effect of BRJ and other supplements such as caffeine [21,26] and phosphate [20]. Therefore, there are a limited number of studies that have assessed the possible ergogenic effect of BRJ on team-sport athletes, such as in female athletes in rugby. In a recent study that aimed to analyze the prevalence of SS consumption in rugby players, a lower consumption of SS in female compared to male rugby players was reported (females: 49% vs. males: 77%) [27], and none of the female rugby players declared consumption of BRJ [27]. Based on the possible ergogenic effect of BRJ, we considered it relevant to study the effects of this ergogenic aid on the neuromuscular performance in female rugby players, similar to the side effects reported with BRJ ingestion. Thus, the aim of this study was to analyze the effect of an acute BRJ supplementation on a neuromuscular battery on a sample of female competitive rugby players.

## 2. Materials and Methods

### 2.1. Participants

Fourteen semi-professional female field rugby players between 18 and 40 years old from the First Spanish Division were recruited for the present study (clinicaltrials.gov (accessed on 8 November 2022); ID: NCT05209126). Exclusion criteria were; suffering from any chronic pathology or an injury in the month prior to the investigation, intolerance to vegetables with high nitrate content (e.g., lettuce, beetroot) and the use of medicines or dietary supplements during the study (e.g., caffeine). Seven participants were tested during the follicular phase of their menstrual cycle, and seven were tested during the luteal phase according to a mobile application (Mycalendar; Period Tracker, Singapore) that identifies main events occurring during the menstrual cycle. Four participants were excluded from the study, who did not attend the second day testing for unspecified reasons. A final sample of fourteen rugby players participated in this study [28]. After being fully informed of the experimental protocols, all female players gave their informed written consent to participate. This study complied with the Declaration of Helsinki and was approved by a Bioethics Commission (code: UI1-PI002).

### 2.2. Experimental Protocol

Two days before each experimental session, dietary NO_3_^−^ intake was restricted by providing subjects with a list of NO_3_^−^ rich foods (e.g., beetroot, celery or spinach) that they should avoid [29]. Participants were encouraged to avoid brushing their teeth or using any oral antiseptic rinse or chewing gum or ingesting sweets that could alter their oral microbiota and interfere with NO_3_^−^ reduction during the 24 h leading up to each experimental trial [30]. Female participants were instructed to refrain from any type of exercise or the ingestion of caffeine 24-h before the experimental trials and to follow a diet sheet consisting of 60% carbohydrates, 30% fat and 10% proteins that should be replicated during the two days of the study. Additionally, participants were provided with a survey to be filled out the following morning relating to potential symptoms (e.g., gastrointestinal upset, red urine, acid reflux, nausea and other perceived discomfort). This survey included several typical side effects associated with BRJ on a yes/no scale that was used previously to assess the side effects derived from BRJ ingestion [25].

### 2.3. Experimental Design

The study design was a randomized double-blinded and placebo-controlled crossover trial. During each trial, 50% of participants ingested the placebo (PLAC) and 50% ingested BRJ with random assignment to each supplement (Research Randomizer, www.randomizer.org (accessed on 8 November 2022)). After BRJ or PLAC intake, female rugby players underwent two identical testing sessions on two different days separated by one week to allow a full recovery and substance wash-out. Female semi-professional rugby players were allocated to receive a 140 mL dose of BRJ containing 12.8 mmol of NO_3_^−^ (Beet-It-Pro Elite Shot, James White Drinks Ltd., Ipswich, UK) or 140 mL PLAC matched in flavor, appearance, and packaging (0.08 mmol of NO_3_^−^, Beet-It-Pro Elite Shot, James White Drinks Ltd., Ipswich, UK) 2.5 hours before each testing session. Prior to the assessment, a familiarization session with all the neuromuscular tests was realized during warm-up. Testing sessions included a neuromuscular test battery consisting of countermovement jump (CMJ), isometric handgrip strength (dominant), 10-m and 30-m sprint test, modified agility *t*-test and the Bronco endurance test. Experimental procedures were performed in a rugby field at the same hour in the evening (20:00 h) to reduce the potential effects of circadian rhythms, as was previously reported in female team-sports disciplines [31]. Environmental conditions were similar between trials (9–12 °C, 45–50% humidity), measured using a portable weather station (Meteorological Station, Künken, Spain), and all the participants performed the same warm-up, as per the usual training protocol.

### 2.4. Countermovement Jump and Dominant Isometric Handgrip Strength

CMJ was performed using a commercially available jump mat (Chronojump. Boscosystems ^®^, Barcelona, Spain). All jumps were initiated from a stationary standing position, followed by a 90° knee flexion and a jump phase [32]. Players were asked to keep their hands on their waist during the entire CMJ and performed two attempts with one minute of rest between trials. The highest value out of two attempts was recorded as the maximum jump, which was used for subsequent statistical analysis. Additional attempts were performed until two consecutive measures differed less than 5%. Furthermore, an isometric handgrip strength test was performed. Two maximum isometric voluntary contractions were measured in the dominant hand using a calibrated handgrip dynamometer (Takei 5101, Tokyo, Japan). Volunteers sat with 0 degrees of shoulder flexion, 0 degrees of elbow flexion, and the forearm and hand in a neutral position [33]. The highest value out of two attempts was recorded as the maximum voluntary handgrip strength. Female rugby players performed two handgrip maximum isometric voluntary contractions with one minute of rest between trials. The maximum value out of two attempts was recorded as the maximum voluntary handgrip strength. Additional attempts were performed until two consecutive measures differed less than 5%.

### 2.5. 10 and 30-Meters Sprint and Modified t-Test

In the 10-m and 30-m sprint test, female rugby players ran at maximal speed for 30-metres in a straight line, and the time needed to cover the distance was measured using two photocell gates placed 1 m above the ground. Photocell gates (Polifemo Radio Light, Microgate, Bolzano, Italy) were located at 0-m, 10-m and 30-m, as previously described in other team-sports disciplines [34]. Two attempts were performed, interspersed with 2 min of passive recovery between repetitions. The faster attempt was used for subsequent analysis. The agility *t*-test is used in the assessment of female rugby players [35]; it requires players to move through a modified T-shape circuit to simulate fast movements in short distances. Participants began the test with both of their feet behind the starting point and sprinted forward to cone B and touched its base with the right hand; facing forward and without crossing feet, they shuffled to the left to cone C and touched its base with the left hand. Then, participants shuffled to the right to cone D and touched its base with the right hand. They shuffled back to the left to cone B and touched its base. Finally, female players ran backward as quickly as possible and returned to line A. The best performance out of two repetitions separated by a 2 min recovery period was recorded for subsequent analysis. The time to complete the T-shape circuit was measured using two electronic time sensors (Microgate, Polifemo Radio Light, Bolzano, Italy) set 1-m above the surface and positioned 3-m apart facing each other on either side of the starting line. Participants began each test 1 m behind the starting line, and the timer started when they passed the first gate.

### 2.6. Bronco Endurance Test

This endurance test is widely used in rugby for determining aerobic fitness in rugby players, and it consists of performing a 1200-m running shuttle. Female rugby players that participated in this study were asked to run from the 0 to the 20-m line and back to 0 m; then run to 40 m and back to 0 m; then run to the 60-m line and back to the 0 m line. Cones were placed to clearly identify 0, 20-m, 40-m and 60-m lines. The execution of the 20 m–40 m–60 m shuttle is considered one repetition. Athletes had to complete five consecutive repetitions as fast as possible to finish the test [36]. The test was measured using electronic photocells (Polifemo Radio Light, Microgate, Bolzano, Italy) located at 0-m, and one repetition was realized and used for the subsequent statistical analysis

### 2.7. Rate of Perception Effort and Side Effects Questionnarie

Participants were asked immediately after completing the neuromuscular battery to provide the rate of perception effort (i.e., 6 to 20 RPE scale of Borg) [37]. Accordingly, participants were first asked to report RPE regarding muscular pain felt in the legs (RPEmuscular); second, participants were asked to report RPE only at the cardiorespiratory level (RPEcardio); and finally, participants had to state global RPE (RPEgeneral), which included features from both the muscular and cardiorespiratory dimensions [38]. In addition, the morning after each of the experimental conditions, the participants filled out a questionnaire based on the main side effects reported (e.g., nausea, reflux, gastrointestinal discomfort) with BRJ ingestion in the hours after the development of the neuromuscular battery [25].

### 2.8. Statistical Analysis

Data are presented as mean ± standard deviation (SD). The Shapiro–Wilk test revealed that data were normally distributed. Paired *t*-tests were performed to compare neuromuscular performance values between BRJ and PLAC trials and statistical significance was set at *p* < 0.05. McNemar’s test was also obtained to detect differences in the prevalence of side effects reported by the questionnaire. Cohen’s d (±95% confidence intervals) for *t*-tests was calculated to estimate the effect size, considering whether it was trivial (<0.19), small (0.20–0.49), medium (0.50–0.79) or large (>0.80) [39]. Calculations were made using SPSS software (version 24, IBM, Armonk, NY, USA), and figures were realized using Graph Prism software (version 8.0.1, GraphPad Software, Inc., San Diego, TX, USA).

## 3. Results

Fourteen semi-professional female rugby players (age: 25.0 ± 3.7 years; body mass: 61.0 ± 6.3 kg; height: 1.64 ± 0.1 m; training volume: 5.7 ± 1.7 h per week) took part in this study. The study blinding was successful, with only 42.9% of the participants (6/14 participants) correctly identifying the supplement that they were receiving.

### 3.1. Countermovement Jump and Isometric Handgrip Strength

In comparison to the PLAC, statistical differences were reported for acute BRJ ingestion in CMJ height (7.7 ± 11.6%; *p* = 0.029; ES = 0.62 (0.42; 0.82) (Figure 1A), but not in dominant isometric handgrip strength (−1.7 ± 6.2%; *p* = 0.274; ES = −0.20 (−0.40; 0.00) (Figure 1B).

### 3.2. 10 and 30-Meters Sprint, Modified t-Test and Bronco Endurance Test

No statistical differences were reported in the 10-m sprint (0.9 ± 5.3%; *p* = 0.588; ES = 0.18 (−0.02; 0.38) (Figure 1C) and 30-m sprint (0.5 ± 2.3%; *p* = 0.441; ES = 0.03 (−0.17; 0.23) (Figure 1D) for BRJ vs. PLAC conditions. Likewise, no statistical differences were reported in the modified *t*-test (−0.6 ± 3.3%; *p* = 0.503; ES = −0.12 (−0.32; 0.08) (Figure 1E) and Bronco Endurance Test (1.9 ± 4.6%; *p* = 0.143; ES = 0.16 (−0.04; 0.35) (Figure 1F).

### 3.3. Rate of Perception Effort and Side Effects Questionnaire

No statistical differences between PLAC and BRJ conditions were reported in RPEmuscular (14.0 ± 1.8 vs. 14.6 ± 2.8 points; *p* = 0.357, ES = 0.26 (0.06; 0.46), RPEcardio (15.7 ± 1.9 vs. 15.0 ± 2.4 points; *p* = 0.526; ES = −0.32 (0.12; 0.52) and RPEgeneral (15.8 ± 2.0 vs. 16.2 ± 1.7 points; *p* = 0.418; ES = 0.22 (0.02; 0.42). According to the side effects reported by the questionnaire regarding BRJ or PLAC ingestion, statistical differences were reported in gastrointestinal upset (*p* = 0.031) with no statistical differences observed in the rest of the scale items (*p* = 0.142–1.000) (Table 1).

## 4. Discussion

The main results of this study showed an enhanced neuromuscular performance assessed by the maximum height reached in a CMJ after BRJ ingestion, but no differences were reported in dominant isometric handgrip strength, linear sprint (10-m and 30-m sprint), modified agility *t*-test or Bronco test. In addition, no differences were reported in rate of perception scales, whereas the side effects questionnaire reported higher gastrointestinal upset symptoms after BRJ compared to PLAC.

Previous to this investigation, only one study has assessed the effect of BRJ on CMJ performance in female trained athletes [23]. The magnitude of these changes in CMJ performance is similar (+7.7%) to the enhancement reported by Jurado-Castro et al. [23] (+6.0%) in a sample of females experienced in resistance training programs. CMJ execution is characterized by a higher rate of force development along brief contraction times. Therefore, height reached in CMJ is considered an indicator of lower limb muscular power [40]. In addition, CMJ is a test sensible for detecting changes caused by SS in athletes [41], and it is used for quantifying fatigue [42] or monitoring training load during a block of training in rugby players [43]. CMJ involves numerous components that include concentric, eccentric and stretch–shortening cycles (SSC) [44]

In female resistance-trained athletes, an enhanced CMJ performance with a higher mean velocity and power output during the execution of a dynamic back squat with a load corresponding to 50% 1RM was reported, while no differences were reported at 75% 1RM [23], whereas in a sample composed by male and female participants, an enhanced power was registered only at a high angular velocity (6.28 rad/s) during isokinetic knee extension [45]. In addition, in moderately trained males, an enhanced power output was reported during a back squat using a flywheel device that involved eccentric and SSC [46]. Nevertheless, studies that have assessed power output during a dynamic back squat with a pause of 2 s between eccentric and concentric phases [47] or low or moderate speed in isokinetic knee extensions [48,49,50] failed to increase performance after BRJ supplementation. In a study, it was reported that, in humans, BRJ increases force production only during the first 50 ms during twitch contractions evoked by supramaximal nerve stimulation [51]. Additionally, in mice, muscle contraction function in type II muscle fiber was improved after NO_3_^−^ supplementation, in addition to an increased content of the calcium-handling proteins, dihydropyridine receptor (DHPR) and calsecuestrin (CASQ) [52]. An increased DHPR and CASQ could increase calcium bioavailability in the myoplasm with a number and velocity of crossed bridges that could potentiate muscle contractions in actions that involve SSC. Eccentric muscle contractions and high velocity contractions, as sport actions that involve SSC, increase the recruitment of type II muscle fibers [53]. Con Attending that BRJ effects are specific for type II muscle fiber contraction without effect in type I muscle fiber [54], it explains the ergogenic effect of BRJ in female trained athletes in this study in accordance with results previously reported [23], because the enhancement of muscle power during the eccentric phase [46] that could be taken advantage in the concentric phase, which could be enhanced during the first part of the contraction [51] with a higher height reached in the CMJ.

Sprint performance depends on the acceleration and maximum speed capability [55]. However, lower distances are mainly influenced by impulse and acceleration, while increased distances are more influenced by SSC, which explains the increased correlation between CMJ and time in sprints from 5-m to 30-m in female rugby players [56]. Therefore, CMJ was highly correlated with the 30-m sprint test in rugby union players [57]. Thus, it would be reasonable to find an enhancement in the 30-m sprint after BRJ, but no effect after BRJ ingestion was observed in the study. In this sense, an enhanced CMJ is not always followed by an enhancement in specific sport actions influenced by the neuromuscular performance. For example, other studies have reported a higher performance in CMJ but not in sprint capability in rugby after caffeine supplementation [58]. The absence of any effect from BRJ in the modified *t*-test is in accordance with results reported in other sport modalities, such as tennis [59] and basketball [60]. The modified *t*-test is one of the most common change of direction speed tests used in sport [61], including rugby [35]. Nevertheless, COD speed is influenced by numerous motor skills (i.e., proper technique), straight sprinting speed, reactive strength, concentric strength and power, ability to be fast in acceleration and deceleration and right–left muscle imbalance [62] All these variables that are not only neuromuscular parameters could result in a difficult test for detecting differences after BRJ and other SS.

Handgrip strength is one of the best physical parameters for detecting rugby player talent [63,64] In this study, BRJ did not affect isometric handgrip strength, in agreement with previous studies that have not found any effect in tennis [59] and basketball players [60]. A possible explanation could be that progressive recruitment during the execution of this test may imply both type I and type II muscle fibers and, therefore, limit the possible ergogenic effect of BRJ that is specific to type II muscle fibers [54]. Endurance performance is also a factor in performance in rugby [4]. The Bronco test is a specific field test for assessing endurance performance in rugby players that has shown a moderate reliability and a high association with the Yo-Yo Intermittent Recovery Test Level 1 (Yo-Yo IRT L1) [65]. However, BRJ is proposed as an ergogenic aid in endurance performance [13]. The performance in the Bronco test was not different in comparison with the placebo. A meta-analysis reported an increased economy (a factor of performance in endurance efforts [14]) after NO_3_^−^ supplementation for moderate and heavy intensity exercises [15]. Nevertheless, the results of the meta-analysis showed an enhanced time-to-exhaustion test (TTE) after NO_3_^−^ or BRJ supplements but not in time-trial tests (TT) [16,17,66]. However, time-to-exhaustion and time-trial tests are used in the assessment of the endurance capacity; TT presents lower variability [67]. The higher variability of TTE can elicit large changes after interventions [67], because a little increased ability to produce power provokes higher changes in TTE [68]. Our results are in line with the results of the meta-analysis [16,17,66], confirming that BRJ is not an ergogenic effect for increasing TT, but future studies should use TTE protocols in female rugby players.

According to the rate-of-perception effort, no differences were found comparing BRJ vs. placebo conditions in RPEmuscular, RPEcardio or RPE general [29,60]. Our data are in agreement with previous studies that do not report differences between conditions [29,60] Finally, referring to the side effects questionnaire, an increment in gastrointestinal upset was reported in this study that is in agreement with previous studies [25]. However, this finding remains controversial since some studies did not report any side effects between conditions [29]. Differences could be attributed to the different beetroot juice doses in the studies (140 vs. 70 mL), differences in sex (female vs. males) or training experience.

## 5. Limitations

First, although menstrual phase was tracked for all participants, the specific control of the menstrual phase was recognized as a limitation of the study. Future research should assess whether the impact of beetroot supplementation varies according to the menstrual cycle phase. Second, future investigations should also be conducted to determine the effect of beetroot juice ingestion in a real competitive rugby context, including a multiday supplementation protocol or a higher dosage than used in the present study. Third, we did not obtain blood or saliva samples, and thus, we were unable to assess the levels of circulating NO_3_^−^ and NO_2_^−^ after beetroot juice administration.

## 6. Conclusions

The results of the present study showed that BRJ ingestion could improve neuromuscular performance in the CMJ test, while no differences in sprint (10-m and 30-m sprint test), agility, isometric handgrip strength and endurance performance (i.e., Bronco Test) were reported. Furthermore, it is worth mentioning that BRJ ingestion could increase the side effects reported (i.e., gastrointestinal upset). Thus, futures studies should be realized to warrant our preliminary findings.

## Figures and Tables

**Figure 1 foods-11-03614-f001:**
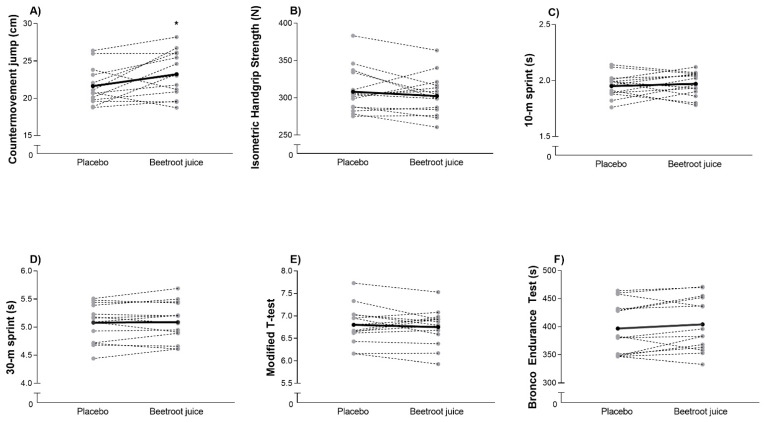
Height during a countermovement jump (**A**), Dominant isometric handgrip strength (**B**), Time during a 10-m sprint test (**C**) and during a 30-m sprint test (**D**), Time for completing agility *t*-test (**E**) and time to complete Bronco endurance test (**F**) after beetroot juice or placebo ingestion in semi-professional female rugby players. Means are represented by a horizontal black bar and each individual is represented by a discontinuous grey line. * Statistical differences were set at *p* < 0.05.

**Table 1 foods-11-03614-t001:** Prevalence of side effects after the ingestion of 140 mL of beetroot juice or 140 mL placebo in semi-professional female rugby players.

Items	Placebo	Beetroot Juice	*p*
Gastrointestinal upset (%)	0 (0%)	4 (28.6%)	0.031 *
Red urine (%)	5 (35.7%)	6 (42.9%)	0.699
Gastric reflux (%)	0 (0%)	1 (7.1%)	0.309
Mild nausea (%)	0 (0%)	2 (14.3%)	0.142
Muscle pain (%)	0 (0%)	0 (0%)	1.000
Headache (%)	0 (0%)	0 (0%)	1.000
Increased excretion of urine (%)	2 (14.3%)	4 (28.6%)	0.357
Increased fatigue (%)	0 (0%)	1 (7.1%)	0.309
Nervousness (%)	0 (0%)	1 (7.1%)	0.309

Data are percentage of affirmative responses to each of the side-effect for 14 semi-professional female rugby players. * Statistical differences (*p* ≤ 0.05).

## Data Availability

All the data are guarded by the corresponding author. Any additional data can be requested from the corresponding author.

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
