# Peer review of "Influence of Beetroot Juice Ingestion on Neuromuscular Performance on Semi-Professional Female Rugby Players: A Randomized, Double-Blind, Placebo-Controlled Study"

_foods, 2022, doi:10.3390/foods11223614_

Round 1

Reviewer 1 Report

Congratulations for the paper.

Everything is fine, but the side effects. They appear in the title so that makes us think they are going to be deeply investigated. Then, they are not mentioned in the introduction and the questionnaire seems rather simple. then, the results to back the side effects are just 1 item. I believe that the side effects should be mentioned, and that it is a very important issue in a practical way, but they are not alligned with the other results. 

My suggestion is that you describe it better and mention why you think this is an important variable in the introduction.

Once again, congratulations for your work.   

ps. there is a spelling error at line 89  " rugby players y no consumption" 

Author Response

Response to Reviewers’ comments

FOODS- 1910846

Influence of beetroot juice ingestion on neuromuscular performance in semi-professional female rugby players: A randomized, double-blind, placebo-controlled study.

We sincerely thank the expert Reviewers and the Editor Foods Journal for their helpful and constructive comments, and for carefully reviewing the manuscript. In this rebuttal letter, we have addressed all the points raised by the expert Reviewers and we have included all the changes proposed throughout the manuscript, highlighted in red. We believe that the comments and suggested changes have significantly improved the quality of this manuscript.

Reviewers' comments:

Reviewer #1:

Congratulations for the paper.

Response: Many thanks for your nice words.

Everything is fine, but the side effects. They appear in the title so that makes us think they are going to be deeply investigated. Then, they are not mentioned in the introduction and the questionnaire seems rather simple. then, the results to back the side effects are just 1 item. I believe that the side effects should be mentioned, and that it is a very important issue in a practical way, but they are not alligned with the other results.

Response: We are agreeing with the reviewer opinion that side effects are not deeply investigated in our research (are only a secondary variable measured). Thus, we have modified manuscript title by “Influence of beetroot juice ingestion on neuromuscular performance on semi-professional female rugby players: A randomized, double-blind, placebo-controlled study” that in our opinion reflects better our investigation.

My suggestion is that you describe it better and mention why you think this is an important variable in the introduction.

Response: According to your suggestions we have included a brief paragraph about side effects “as same as side effects reported with the beetroot juice ingestion on different team-sports athletes.” (Lines 88-89).

  1. there is a spelling error at line 89 " rugby players y no consumption"

Response: Fixed. We have modified by “consumption has been reported by this team”

Reviewer 2 Report

Nutritional supplements are used by athletes to maximize the benefits of their training and increase their athletic performance. Nitric oxide (NO), which has several roles in improved blood flow, gas exchange, mitochondrial biogenesis and efficiency, and strengthened muscular contraction, is boosted by beetroot juice that contains NO3-. Despite inconsistent data in the literature, some research' findings hypothesize that nitrate ingestion could increase intermittent type exercise performance in recreational athletes.

The authors present the current paper focusing on the ingestion of beetroot juice in female rugby players and their neuromuscular effects and side effects to ingestion. The article is well structured but fails to justify the chosen subject, if we analyze the contemporary specialized literature, for several reasons:

First of all, the article it is very similar to other profile articles, adding to those that specify that beetroot juice has in general no effect on athletes, which does not bring added scientific value. The very first author of the current author, Álvaro López, also wrote 2 years ago the article "Does Acute Beetroot Juice Supplementation Improve Neuromuscular Performance and Match Activity in Young Basketball Players? A Randomized, Placebo-Controlled Study, - Álvaro López-Samanes, et al." where they basically used Young Basketball Players instead of female rugby and has a similarity in structure and information of over 60% -reference 60. More, the article concluded that it has no effect. The present article just exploits the fact that they haven't been tested enough on women and also has High similarities with reference no.23 “Acute Effects of Beetroot Juice Supplements on Lower-Body Strength in Female Athletes: Double-Blind Crossover Randomized Trial”.

Secondly, many of the similar articles do studies on repeated ingestion over 6-7 days to allow the body to assimilate and get used to beet root juice. In this article, the authors offer the ingredients to the participants only a few hours before the test, once, after which they pass them through a battery test, repeating the event in a week. There are too many variables, the human body is too complex, and the sample of participants is too small to be relevant and be able to generate a conclusion considered scientific like in the title “beetroot juice improves neuromuscular performance…in female rugby players”. At best it could be used “in some” or at least do not mention it so obviously in the title, since just one of the six tests had incremental increased results. The variables previously mentioned were used by the authors to indicate some inconsistencies between the results (which in theory should have been) and just some correlations were made. But, although it sounds scientific, since we are also talking about females, everything gets complicated. The authors specify in the procedures that seven participants were tested during the follicular phase of their menstrual cycle, and seven were tested during the luteal phase. Furthermore, they do not elaborate on this aspect even if it is relevant, for such a small number of tests, or about the fact that the cycle significantly influences the performance. Practically, from only 14 subjects, tested one week apart, there are high chances that the results were also influenced by the "period" in which they were found. The menstrual cycle has a follicular and a luteal phase. The follicular phase is split in early (first week) and late follicular phases (second week), and the luteal phase is split into 3 phases – early, mid and late – luteal, each taking between 4 and 5 days. In the late follicular estrogen rises to a critical point, there is increased secretion of gonadotropin releasing hormone, and the mid luteal phase contains the peak in progesterone and the second smaller peak in estrogen, and those can influence overall sport performance by both perceived and physical factors, as emerged from the work listed in the bibliography position 28, "Janse de Jonge, X.A. Effects of the menstrual cycle on exercise performance. Sports Med 2003, 33, 833-851, 465 doi:10.2165/00007256-200333110-00004".

To this extent, without denigrating the work done by the authors, one could skip the BRJ addition, and can generate similar conclusions, following the same procedures, by creating connections with the hormone, or one could replace BRJ with a bar of snickers or any other fruit juice and develop new connections with the composition of the latter, positive or negative. This because, in fact, we are talking about replacing an element from the menu of some athletes, one time, following the performance immediately after and generating a conclusion considering that all the other physical and mental components of the very small number of participants (over 1.5 million women in rugby) is fixed and known, as a modification of a parameter of a machine, which is not the case with the human body, which anyway in the training regime also has incremental peaks of performance increases, generated by the repetition of training.

Thirdly, BR is used as a supplement, but sold worldwide for providing the support for athletic performance, support for blood pressure, for immune system, and a healthy digestive system in packages that contain dozens of capsules that are taken during long consecutive sports training periods, not just one before a match. Thus, they could end up increasing plasma and salivary nitrate and nitrite concentrations in some cases for example or have significant side effects on the digestion system

Proofreading suggestions for authors

·         Line 21 - please remove the first "sports" word since it is repetead again after 2 words

·         Line 21 - please correct to "separated by one week between protocols"

·         Keywords: Football? – could the authors please justify why football is inserted in the keywords since the article is about rugby

·         Line 68 – please remove the second “BRJ” from the text since is previously mentioned in the beginning of the line

·         Line 69 – please ad punctuation (.) between sentences

·         Line 70 - please remove the second “BRJ” from the text since is previously mentioned in the beginning of the sentence and avoid so many repetitions

·         Line 84 - please remove double punctuation (.) between sentences

·         Line 89 - please remove y or correct it

·         Line – 96 - please use : instead of ;

·         Table 1 – could the authors explain why is red urine even in placebo taken by semi-professional female rugby players in high amounts

·         Line 281 – please remove double space

·         Line 287 – please correct to “a high rate” or “high rates”

·         Line 295 – please choose the correct form “50% 1 RM, but not at 75% 1RM”

·         Line 319 – please correct “theacceleration” to „the acceleration”

·         Line 323 - Thus, it would be reasonable to find an enhancement in the 30-m sprint after BRJ, but  no effect after BRJ ingestion was observed in the study – since it was not found, where is the justification for the title of the article

·         Line 331-333- since there are so many variables, could it be possible that the only result influenced by the BRJ to be just a "happening" since the majority did not

·         Line 341 – please ad punctuation (.) between sentences

·         Line 357 – please ad (,)

·         Line 371-373 – please rephrase or correct the prepositions since in order to create sense for the reader

Author Response

Response to Reviewers’ comments

FOODS- 1910846

Influence of beetroot juice ingestion on neuromuscular performance in semi-professional female rugby players: A randomized, double-blind, placebo-controlled study.

We sincerely thank the expert Reviewers and the Editor Foods Journal for their helpful and constructive comments, and for carefully reviewing the manuscript. In this rebuttal letter, we have addressed all the points raised by the expert Reviewers and we have included all the changes proposed throughout the manuscript, highlighted in red. We believe that the comments and suggested changes have significantly improved the quality of this manuscript.

Reviewers' comments:

Reviewer #2:

Nutritional supplements are used by athletes to maximize the benefits of their training and increase their athletic performance. Nitric oxide (NO), which has several roles in improved blood flow, gas exchange, mitochondrial biogenesis and efficiency, and strengthened muscular contraction, is boosted by beetroot juice that contains NO3-. Despite inconsistent data in the literature, some research' findings hypothesize that nitrate ingestion could increase intermittent type exercise performance in recreational athletes. The authors present the current paper focusing on the ingestion of beetroot juice in female rugby players and their neuromuscular effects and side effects to ingestion. The article is well structured but fails to justify the chosen subject, if we analyze the contemporary specialized literature, for several reasons: First of all, the article it is very similar to other profile articles, adding to those that specify that beetroot juice has in general no effect on athletes, which does not bring added scientific value. The very first author of the current author, Álvaro López, also wrote 2 years ago the article "Does Acute Beetroot Juice Supplementation Improve Neuromuscular Performance and Match Activity in Young Basketball Players? A Randomized, Placebo-Controlled Study, - Álvaro López-Samanes, et al." where they basically used Young Basketball Players instead of female rugby and has a similarity in structure and information of over 60% -reference 60. More, the article concluded that it has no effect. The present article just exploits the fact that they haven't been tested enough on women and also has High similarities with reference no.23 “Acute Effects of Beetroot Juice Supplements on Lower-Body Strength in Female Athletes: Double-Blind Crossover Randomized Trial”.

Response: According to the author response, we would like to mention that according to our knowledge this is the first study that have analysed the influence of beetroot juice ingestion on non-aquatic female team sports athletes. In addition, according to a previous review (Wickham et al. 2019) that established that most of the scientific studies that have studied beetroot juice topic did not cover female population. Thus, in our humble opinion, this article could increase the knowledge about this gap on the scientific literature and add useful information about the effects of this dietary supplement in team-sports womens athletes.

Reference: Wickham KA, Spriet LL. No longer beeting around the bush: a review of potential sex differences with dietary nitrate supplementation. Appl Physiol Nutr Metab. 2019;44(9):915–24

Secondly, many of the similar articles do studies on repeated ingestion over 6-7 days to allow the body to assimilate and get used to beet root juice. In this article, the authors offer the ingredients to the participants only a few hours before the test, once, after which they pass them through a battery test, repeating the event in a week. There are too many variables, the human body is too complex, and the sample of participants is too small to be relevant and be able to generate a conclusion considered scientific like in the title “beetroot juice improves neuromuscular performance…in female rugby players”. At best it could be used “in some” or at least do not mention it so obviously in the title, since just one of the six tests had incremental increased results. The variables previously mentioned were used by the authors to indicate some inconsistencies between the results (which in theory should have been) and just some correlations were made. But, although it sounds scientific, since we are also talking about females, everything gets complicated. The authors specify in the procedures that seven participants were tested during the follicular phase of their menstrual cycle, and seven were tested during the luteal phase. Furthermore, they do not elaborate on this aspect even if it is relevant, for such a small number of tests, or about the fact that the cycle significantly influences the performance. Practically, from only 14 subjects, tested one week apart, there are high chances that the results were also influenced by the "period" in which they were found. The menstrual cycle has a follicular and a luteal phase. The follicular phase is split in early (first week) and late follicular phases (second week), and the luteal phase is split into 3 phases – early, mid and late – luteal, each taking between 4 and 5 days. In the late follicular estrogen rises to a critical point, there is increased secretion of gonadotropin releasing hormone, and the mid luteal phase contains the peak in progesterone and the second smaller peak in estrogen, and those can influence overall sport performance by both perceived and physical factors, as emerged from the work listed in the bibliography position 28, "Janse de Jonge, X.A. Effects of the menstrual cycle on exercise performance. Sports Med 2003, 33, 833-851, 465 doi:10.2165/00007256-200333110-00004". To this extent, without denigrating the work done by the authors, one could skip the BRJ addition, and can generate similar conclusions, following the same procedures, by creating connections with the hormone, or one could replace BRJ with a bar of snickers or any other fruit juice and develop new connections with the composition of the latter, positive or negative. This because, in fact, we are talking about replacing an element from the menu of some athletes, one time, following the performance immediately after and generating a conclusion considering that all the other physical and mental components of the very small number of participants (over 1.5 million women in rugby) is fixed and known, as a modification of a parameter of a machine, which is not the case with the human body, which anyway in the training regime also has incremental peaks of performance increases, generated by the repetition of training.

Response: We are completely agreeing with the reviewer when he stablished that “human body in too complex”. For this reason during this study we tried to control all the confounding variables that could affects to our results such as; a) dietary NO3- intake (i.e. NO3-rich foods was restricted two days before of each experimental session) (Fernández-Elías JANA 2022), b) avoid brushing their teeth or using any oral antiseptic rinse or chewing gum or ingesting sweets that could alter their oral microbiota (i.e., interfere with NO3- reduction processes) during the 24 h leading up to each experimental trial) (Burleigh NO 2019), c) same meal composition during the previous days of the experimental sessions, d) time-of-day when beetroot intake is ingested (Dumar IJERPH 2021). According to menstrual cycle this study involved inclusion of female field hockey players, we aimed to standardise the logistical operation of the study (i.e. the team were tested during squad availability). As a means to account for potential influence of hormonal fluctuations, players were asked to use Mycalendar to attempt to track which phase they were in when they started. It is challenging to start everyone at the same precise time due to when the squad meet, but this approach at least allowed us to account for this variable (which is recognised as a limitation). In addition, your we are agreeing with you menstrual cycle comments and now we have added in the limitations section “First, although menstrual phase was tracked for all participants, however, specific control of menstrual phase was recognised as a limitation of the study. Future research should assess whether the impact of beetroot supplementation varies according to menstrual phase” (Lines 349-351)

References:

Burleigh, M.; Liddle, L.; Muggeridge, D.J.; Monaghan, C.; Sculthorpe, N.; Butcher, J.; Henriquez, F.; Easton, C. Dietary nitrate supplementation alters the oral microbiome but does not improve the vascular responses to an acute nitrate dose. Nitric Oxide 2019, 89, 54-63, doi:10.1016/j.niox.2019.04.010.

Dumar, A.M.; Huntington, A.F.; Rogers, R.R.; Kopec, T.J.;Williams, T.D.; Ballmann, C.G. Acute Beetroot juice supplementation attenuates morning-associated decrements in supramaximal exercise performance in trained sprinters. Int. J. Environ. Res. Public Health 2021, 18, 412.

Fernandez-Elias, V.; Courel-Ibanez, J.; Perez-Lopez, A.; Jodra, P.; Moreno-Perez, V.; Coso, J.D.; Lopez-Samanes, A. Acute Beetroot Juice Supplementation Does Not Improve Match-Play Activity in Professional Tennis Players. J Am Nutr Assoc 2022, 41, 30-37, doi:10.1080/07315724.2020.1835585

Thirdly, BR is used as a supplement, but sold worldwide for providing the support for athletic performance, support for blood pressure, for immune system, and a healthy digestive system in packages that contain dozens of capsules that are taken during long consecutive sports training periods, not just one before a match. Thus, they could end up increasing plasma and salivary nitrate and nitrite concentrations in some cases for example or have significant side effects on the digestion system

Response: We are agreeing with the reviewer, and we have included that we did not measure plasma or salivary nitrate and nitrite concentrations in the limitations section “Third, we did not obtain blood or saliva samples and thus, we were unable to assess the levels of circulating NO3- and NO2- after beetroot juice administration” (lines 354-356). In addition, as the reviewer mentioned we measures side-effects for expending the knowledge about the effects of beetroot juice on digestion system among others.

Proofreading suggestions for authors

  • Line 21 - please remove the first "sports" word since it is repetead again after 2 words

Response: Done. Thanks.

  • Line 21 - please correct to "separated by one week between protocols"

Response: Corrected as suggested by the reviewer.

  • Keywords: Football? – could the authors please justify why football is inserted in the keywords since the article is about rugby

Response: Nice appreciation. We have modified keywords.

  • Line 68 – please remove the second “BRJ” from the text since is previously mentioned in the beginning of the line

Response: Fixed.

  • Line 69 – please ad punctuation (.) between sentences

Response: Done. Thanks.

  • Line 70 - please remove the second “BRJ” from the text since is previously mentioned in the beginning of the sentence and avoid so many repetitions

Response: Corrected as suggested by the reviewer.

  • Line 84 - please remove double punctuation (.) between sentences

Response: Good point. We have removed.

  • Line 89 - please remove y or correct it

Response: Fixed. Thanks.

  • Table 1 – could the authors explain why is red urine even in placebo taken by semi-professional female rugby players in high amounts

Response: Many thanks for this insightful question. Beetroot juice or nitrate-depleted beetroot juice placebo were matched in flavour, appearance, packaging and composition (only differences between treatments were reported in the nitrate’s concentration (12.8 vs 0.04 mmol of NO3-). Thus, the substance that converted to red urine (betalain) is presented in both products.

  • Line 281 – please remove double space

Response: Corrected as suggested by the reviewer.

  • Line 287 – please correct to “a high rate” or “high rates”

Response: Done. Thanks.

  • Line 295 – please choose the correct form “50% 1 RM, but not at 75% 1RM”

Response: Change by “to 50% 1RM while no differences reported at 75% 1RM”.

  • Line 319 – please correct “theacceleration” to „the acceleration”

Response: Corrected as suggested by the reviewer.

  • Line 323 - Thus, it would be reasonable to find an enhancement in the 30-m sprint after BRJ, but no effect after BRJ ingestion was observed in the study – since it was not found, where is the justification for the title of the article

Response: Nice appreciation. According to the reviewer opinion we have changed the article title by “Influence of beetroot juice ingestion on neuromuscular performance on semi-professional female rugby players: A randomized, double-blind, placebo-controlled study”

  • Line 331-333- since there are so many variables, could it be possible that the only result influenced by the BRJ to be just a "happening" since the majority did not

Response: The expectation is that height increment reported in CMJ will be followed for an enhancement in acceleration and velocity in short distance measured in our study. However, our results are in agreement with previous studies that reported changes in CMJ, but not in sprint performance in rugby players (Ranchordas et al., 2019). It is worth to mention that sprint training usually presented very little changes during sport specific intervention, while jump height is more sensible tool for detecting neuromuscular changes after a training program (Gathercole et al. 2015) Thus, in our humble opinion velocity in sports distance could be less sensitive for detecting neuromuscular differences after nutritional comparing to CMJ that could explain the findings obtained in our study

References:

Gathercole R, Sporer B, Stellingwerff T, Sleivert G.Int J Sports Physiol Perform. 2015 Jan;10(1):84-92. doi: 10.1123/ijspp.2013-0413.

Ranchordas, M.K.; Pratt, H.; Parsons, M.; Parry, A.; Boyd, C.; Lynn, A. Effect of caffeinated gum on a battery of rugby-specific tests in trained university-standard male rugby union players. J Int Soc Sports Nutr 2019, 16, 17, doi:10.1186/s12970-019-0286-7.

  • Line 341 – please ad punctuation (.) between sentences

Response: Fixed. Thanks.

  • Line 357 – please ad (,)

Response: Corrected as suggest by the reviewer.

  • Line 371-373 – please rephrase or correct the prepositions since in order to create sense for the reader

Response: Based on the suggestions of the reviewers, conclusions have been revised.

Reviewer 3 Report

The sample sizes - fourteen semi-professional female field rugby players between 18 and 40 years old partitioned into two equal groups. placebo and injection BRJ - are extremely small to anticipate some significant and important results.

The authors have to justify their choice of such small sample.  

Placebo is not described.

Manuscript contains contradictory statements. On the one hand, it is declared that “the study design was a randomized double-blinded and placebo-controlled crossover trial.” On the other hand “the study blinding was successful with only the 42.9% of the participants (6/14 participants) correctly identifying the supplement that they were receiving.”

Randomization and blinding are obligative procedures in the clinical trials. So only the results with these 6 participants should be considered.

Data in Table 1 is presented incorrectly: total number of side effect (18) is larger then the number of participants. This means that the “comorbidity” of the side effects was observed and the data should be presented as the observed numbers and proportions of all possible side effect combinations.

Conclusion is also very contradictory: on the one hand, BRJ ingestion improves CMJ, but at the same time it provokes much more side effects than in PLAC. Consumers of ingestion of BRJ could improve CMJ but will receive several discomfort conditions.

Thus, it is difficult to understand do authors recommend ingestion BRJ or not.

Author Response

Response to Reviewers’ comments

FOODS- 1910846

Influence of beetroot juice ingestion on neuromuscular performance in semi-professional female rugby players: A randomized, double-blind, placebo-controlled study.

We sincerely thank the expert Reviewers and the Editor Foods Journal for their helpful and constructive comments, and for carefully reviewing the manuscript. In this rebuttal letter, we have addressed all the points raised by the expert Reviewers and we have included all the changes proposed throughout the manuscript, highlighted in red. We believe that the comments and suggested changes have significantly improved the quality of this manuscript.

Reviewers' comments:

Reviewer #3:

The sample sizes - fourteen semi-professional female field rugby players between 18 and 40 years old partitioned into two equal groups. placebo and injection BRJ - are extremely small to anticipate some significant and important results. The authors have to justify their choice of such small sample. 

Response: Nice point. We have analysed the sample size required determined by statistical power calculation based on previous studies (Clifford et al. 2016). The minimum number of participants required to detect an 8 ± 6 % difference in counter movement jump performance between two groups, with a power of 0.80 and two-tailed α level set at 0.05 was estimated as seven per group using G*Power software (v. 3.1.9, Düsseldorf, Germany). In addition, to recruit well-trained female participants is always a huge challenge in the sports context field.

References:

Clifford T, Bell O, West DJ, Howatson G, Stevenson EJ (2016) The effects of beetroot juice supplementation on indices of muscle damage following eccentric exercise. Eur J Appl Physiol 116 (2):353-362.

Placebo is not described.

Response: Nice appreciation. We have included in the manuscript “or 140 mL PLAC matched in flavour, appearance, and packaging (0.08 mmol of NO3-, Beet-It-Pro Elite Shot, James White Drinks Ltd., Ipswich, UK) 2 ½ hours before each testing session”.

Manuscript contains contradictory statements. On the one hand, it is declared that “the study design was a randomized double-blinded and placebo-controlled crossover trial.” On the other hand “the study blinding was successful with only the 42.9% of the participants (6/14 participants) correctly identifying the supplement that they were receiving.”. Randomization and blinding are obligative procedures in the clinical trials. So only the results with these 6 participants should be considered.

Response: Our study was a randomized double-blinded and placebo-controlled crossover trial, where all the female rugby players involved in this study received both treatments (beetroot juice or placebo) with one 1-week protocols (for reducing the influence on training load among others). In addition, such as frequently used and recommended in the sports nutrition studies we analyse the effectivity of guessing (Del Coso et al 2014) detecting that less than 50% of our sample size correctly detecting the treatments that received each day. Thus, we can conclude the guess effectivity.

Reference:

Salinero JJ, Lara B, Abian-Vicen J, Gonzalez-Millán C, Areces F, Gallo-Salazar C, Ruiz-Vicente D, Del Coso J. The use of energy drinks in sport: perceived ergogenicity and side effects in male and female athletes. Br J Nutr. 2014 Nov 14;112(9):1494-502.

Data in Table 1 is presented incorrectly: total number of side effect (18) is larger then the number of participants. This means that the “comorbidity” of the side effects was observed, and the data should be presented as the observed numbers and proportions of all possible side effect combinations.

Response: Nice question. However, when applied a side-effects questionnaire some of the participants reported more than one secondary effect such as previously reported in beetroot juice studies (Wickham et al. 2019).

Reference:

Wickham KA, McCarthy DG, Pereira JM, Cervone DT, Verdijk LB, van Loon LJC, Power GA, Spriet LL. No effect of beetroot juice supplementation on exercise economy and performance in recreationally active females despite increased torque production. Physiol Rep. 2019 Jan;7(2):e13982.

Conclusion is also very contradictory: on the one hand, BRJ ingestion improves CMJ, but at the same time it provokes much more side effects than in PLAC. Consumers of ingestion of BRJ could improve CMJ but will receive several discomfort conditions.

Response: Good point. We have modified the answer by “The results of the present study showed that BRJ ingestion could improve neuromuscular performance in CMJ test, while no differences in sprint (10-m and 30-m sprint test), agility, isometric handgrip strength and endurance performance (i.e., Bronco Test) were reported. Therefore, it is  worth to mention that BRJ ingestion could increase side effects reported (i.e., gastrointestinal upset).

Thus, it is difficult to understand do authors recommend ingestion BRJ or not.

Response: We are agree with the reviewer and we have rewording the conclusion by BRJ ingestion could improve CMJ performance while no improvements were reported in isometric handgrip strength/endurance performance/agility/velocity in short distances. Due in our knowledge this is the first study realized in female team sports (i.e., non-aquatic sports) covering this topic futures studies should be realized for warranted our preliminary findings.

Round 2

Reviewer 2 Report

The authors have worked hard and enhanced the papers quality by addressing all the issues presented (mainly grammar) and changed the title so that it reflects the results presented. They added the limitation section that further impacts the interpretation of the findings and gives more accurate understanding, although niched. I have no other negative aspects to present.

Reviewer 3 Report

In the abstract the authors’ conclusion is: Ingestion of BRJ improves neuromuscular performance (CMJ) while no differences were reported in the rest of variables measured the neuromuscular battery.

However, at the end of the paper conclusion sounds more uncertain: The results of the present study showed that BRJ ingestion could improve neuromuscular performance in CMJ test while no differences in sprint (10-m and 30-m sprint test), agility, isometric handgrip strength and endurance performance (i.e., Bronco Test) were reported.

CMJ test was only one of five tests, so adjustment for the multiplicity should be applied, and real (adjusted for the multiplicity) p-value becomes insignificant: p[adj] = 0.029×5 = 0.145.

Moreover, they mention: BRJ ingestion could increase side effects reported (i.e., gastrointestinal upset). Really, the rate of side effect appeared to be significantly higher in the experimental group then in the control group. And this fact calls into question the feasibility of further research.

Two compered groups were independent, nonetheless the authors applied paired t-test. This is a serious unacceptable mistake.

The authors used an outdated interpretation of the obtained values of Cohen’s effect size (ES): trivial (< 0.19), small (0.20–0.49), medium (0.50–0.79), or large (> 0.80). Modern interpretation used in the sport medicine is: trivial (0 - 0.2), small (0.2 - 0.6), medium (0.6 - 1.2), large (1.2 - 2.0), very large (2.0 - 4.0), nearly perfect (> 0.4).

Hopkins W. A New View of Statistics. 2014. https://www.sportsci.org/resource/stats/newview.html

https://complementarytraining.net/free-will-hopkins-a-new-view-of-statistics-pdf-printout/

Jovanović M. bmbstats: Bootstrap Magnitude-based Statistics for Sports Scientists. 2020.

ISBN: 978-86-900803-5-9

Another serious shortcoming: authors did not provide mandatory confidence limits for the estimates of Cohen’s ES.

The work cannot be improved and must be rejected.

Reported data, results and conclusions do not agree with those reported by the others on the effects of beetroot juice. Some of them are cited below:

1. Berjisian, E., McGawley, K., Saunders, B. et al. Acute effects of beetroot juice and caffeine co-ingestion during a team-sport-specific intermittent exercise test in semi-professional soccer players: a randomized, double-blind, placebo-controlled study. BMC Sports Sci Med Rehabil 14, 52 (2022). https://doi.org/10.1186/s13102-022-00441-1

Beetroot juice (BJ) and caffeine (CAF) are considered as ergogenic aids among athletes to enhance performance, however, the ergogenic effects of BJ and CAF co-ingestion are unclear during team-sport-specific performance. This study aimed to investigate the acute effects of BJ and CAF co-ingestion on team-sport-specific performance, compared with placebo (PL), BJ, and CAF alone.

Hedge’s g effect sizes with a small sample sieze correction and 95% confidence intervals [CIs] were calculated for distance covered during the YYIR1, with minimum threshold values of 0.01, 0.2, 0.5, and 0.8 used to describe effect sizes as very small, small, moderate and large [32]. Results were interpreted according to the statistical probabilities of rejecting the null hypothesis (H0) in the following categories: P ≥ 0.1: no evidence against H0; 0.05 ≤ P < 0.1: weak evidence against H0; 0.01 ≤ P < 0.05: moderate evidence against H0; 0.001 ≤ P < 0.01: strong evidence against H0; P < 0.001: robust evidence against H0 [33].

These results suggest, neither acute co-ingestion of BJ + CAF nor BJ or CAF supplementation alone significantly affected team-sport-specific performance compared to the PL treatment.

Soccer athletes and coaches should be aware that these supplements may not enhance soccer-specific exercise capacity or cognitive function.

2. ValentínE Fernández-ElíasJavier Courel-IbáñezAlberto Pérez-LópezPablo JodraVictor Moreno-PérezJuan Del CosoÁlvaro López-Samanes. Acute Beetroot Juice Supplementation Does Not Improve Match-Play Activity in Professional Tennis Players. J Am Nutr Assoc. 2022 Jan;41(1):30-37. doi: 10.1080/07315724.2020.1835585. 

The current results indicated that acute ingestion of a commercialised shot of nitrate-rich beetroot juice (70 mL containing 6.4 mmol of NO3-) did not produce any performance benefit on tennis match play. Thus, acute beetroot juice supplementation   seems   an   ergogenic aid with little value to enhance physical performance in professional tennis players.

3. López-Samanes Á, Gómez Parra A, Moreno-Pérez V, Courel-Ibáñez J. Does Acute Beetroot Juice Supplementation Improve Neuromuscular Performance and Match Activity in Young Basketball Players? A Randomized, Placebo-Controlled Study. Nutrients. 2020 Jan 9;12(1):188. doi: 10.3390/nu12010188

Acute moderate doses of BJ (12.8 mmol of NO3−) was not effective in improving neuromuscular performance (jump height, isometric handgrip strength, sprint, and agility) or physical match requirements in young trained basketball players the day of the competition. 

4. Naomi M. Cermak, Peter Res, Rudi Stinkens, Jon O. Lundberg, Martin J. Gibala, and Luc J.C. van Loon. No Improvement in Endurance Performance After a Single Dose of Beetroot Juice.  International Journal of Sport Nutrition and Exercise Metabolism, 2012, 22, 470 -478.

Ingestion of a single bolus of concentrated (140 ml) beetroot juice (8.7 mmol NO3 –) does not improve subsequent 1-hr time-trial performance in well-trained cyclists.

5. López-Samanes Á, Pérez-López A, Moreno-Pérez V, Nakamura FY, Acebes-Sánchez J, Quintana-Milla I, Sánchez-Oliver AJ, Moreno-Pérez D, Fernández-Elías VE, Domínguez R. Effects of Beetroot Juice Ingestion on Physical Performance in Highly Competitive Tennis Players. Nutrients. 2020; 12(2):584. https://doi.org/10.3390/nu12020584

Our data suggest that low doses of BJ (70 mL) consumption do not enhance tennis physical performance.

6. Chazelas E, Pierre F, Druesne-Pecollo N, Esseddik Y, Szabo de Edelenyi F, Agaesse C, De Sa A, Lutchia R, Gigandet S, Srour B, Debras C, Huybrechts I, Julia C, Kesse-Guyot E, Allès B, Galan P, Hercberg S, Deschasaux-Tanguy M, Touvier M. Nitrites and nitrates from food additives and natural sources and cancer risk: results from the NutriNet-Santé cohort. Int J Epidemiol. 2022 Aug 10;51(4):1106-1119. doi: 10.1093/ije/dyac046

Food additive nitrates and nitrites were positively associated with breast and prostate cancer risks, respectively. Although these results need confirmation in other large-scale prospective studies, they provide new insights in a context of lively debate around the ban of these additives from the food industry.